# Total Dairy, Cheese and Milk Intake and Arterial Stiffness: A Systematic Review and Meta-analysis of Cross-Sectional Studies

**DOI:** 10.3390/nu11040741

**Published:** 2019-03-29

**Authors:** Ana Diez-Fernández, Celia Álvarez-Bueno, Vicente Martínez-Vizcaíno, Mercedes Sotos-Prieto, José I Recio-Rodríguez, Iván Cavero-Redondo

**Affiliations:** 1Centro de Estudios Socio-Sanitarios, Universidad de Castilla-La Mancha, 16071 Cuenca, Spain; ana.diez@uclm.es (A.D.-F.); Vicente.martinez@uclm.es (V.M.-V.); Ivan.Cavero@uclm.es (I.C.-R.); 2Facultad de Enfermería, Universidad de Castilla-La Mancha, 16071 Cuenca, Spain; 3Facultad de Ciencias de la Salud, Universidad Autónoma de Chile, 1670 Talca, Chile; 4Department of Environmental Health, Harvard TH Chan School of Public health, Harvard Medical School, Boston, MA 02115, USA; sotospri@ohio.edu; 5Department of Food Sciences and Nutrition, School of Applied Health Sciences and Wellness, Ohio University, Athens, OH 45701, USA; 6Diabetes Institute, Ohio University, Athens, OH 45701, USA; 7Institute of Biomedical Research of Salamanca (IBSAL), Primary Health Care Research Unit, La Alamedilla Health Center, Health Service of Castilla y León (SACYL), Primary Care Prevention and Health Promotion Research Network (REDIAPP), 37007 Salamanca, Spain; donrecio@gmail.com; 8Departamento Enfermería y Fisioterapia, Universidad de Salamanca, 37007 Salamanca, Spain

**Keywords:** arterial stiffness, pulse wave velocity, dairy product, milk, meta-analysis, systematic review

## Abstract

The aim of this review was to determine the relationship between dairy product consumption and arterial stiffness, measured by pulse wave velocity (PWV). We systematically searched the Medline, Embase and Web of Science databases until 30 January 2019 for cross-sectional data from studies addressing the association between dairy product consumption and PWV. This study was registered with PROSPERO (CRD42018110528). Both the inverse-variance fixed effects method and the DerSimonian and Laird method were used to compute pooled estimates of effect size (ES) and the respective 95% confidence intervals (CIs). Seven studies were included in the meta-analysis, with a total of 16,443 patients. Total dairy product (ES = −0.03; 95% CI [−0.04, −0.01]) and cheese (ES = −0.04; 95% CI [−0.07, −0.01]) consumption were weak, but significantly associated with lower PWV levels. Conversely, milk intake showed no significant association with PWV (ES = 0.02; 95% CI [−0.01, 0.05]). Heterogeneity in the ES was not important for the three groups of dairy products assessed. This systematic review and meta-analysis of seven studies found no detrimental effects of dairy product consumption on arterial stiffness measured by PWV. Due to the scarcity of studies, further investigations are warranted to clarify the role of dairy products on arterial stiffness.

## 1. Introduction

Cardiovascular disease (CVD) represents the main cause of mortality worldwide [1]. Certain behaviors, such as an unhealthy diet and physical inactivity, have increased in parallel with the prevalence of CVD risk factors, including obesity, hypertension and diabetes. Increasingly, evidence from longitudinal studies consistently supports a negative association between dairy product intake and the risk of CVD, which is in accordance with observational studies that report a negative association between milk intake and obesity [2], diabetes [3] and hypertension [4]. Moreover, a recent meta-analysis of prospective studies have also shown a null or weak inverse effect of dairy products on CVD [5].

Similarly, recent studies have reported that dairy product intake is associated with other CVD risk indicators, such as arterial stiffness [6]. A characteristic of the aging process is the stiffening of large arteries, which may promote the progression of some chronic diseases, particularly hypertension. Conversely, certain behaviors, such as smoking, a sedentary lifestyle or unhealthy diet, and certain cardiometabolic diseases, such as obesity and type 2 diabetes mellitus, are consistently correlated with the progression of arterial stiffness [7,8,9].

Dairy products include milk, cream, butter, cheese, yogurt, frozen desserts and whey, among others. They contain several micro- and macronutrients, including saturated fatty acids (SFAs). Indeed, the SFA content is the main reason why dairy product intake has received an unfavorable reputation in recent years [10]. The literature examining the relationship between dairy product intake and arterial stiffness is scarce and findings are inconsistent, although recent results from the Prospective Urban Rural Epidemiology (PURE) multinational cohort study suggest that dairy consumption might be associated with a lower risk of mortality and major cardiovascular disease events [11]. Moreover, recent meta-synthesis studies have addressed the relationship between milk and dairy consumption and CVD [12], diabetes [13] and all-cause mortality [14]. 

Increased aortic pulse wave velocity (PWV), the most valid and reliable measure of arterial stiffness, has been associated with a higher risk of cardiovascular events and all-cause mortality. A systematic review and meta-analysis [6] reported that the prediction ability of arterial stiffness is higher in subjects at increased risk of cardiovascular disease, estimating a 14% increase in total cardiovascular risk with every 1 m/s increase in PWV. Among the indexes proposed to estimate PWV, carotid-femoral pulse wave velocity (cfPWV) is accepted as the gold standard noninvasive method [15].

This systematic review and meta-analysis aimed to synthesize the evidence of the relationship between the consumption of dairy products and arterial stiffness (measured by PWV).

## 2. Materials and Methods

This systematic review and meta-analysis is reported according to the Preferred Reporting Items for Systematic Reviews and Meta-analyses (PRISMA) guidelines [16] (Figure 1), and follows the recommendations of the Cochrane Collaboration Handbook [17]. The study was also registered through the International Prospective Register of Systematic Reviews (registration number: CRD42018110528).

### 2.1. Data Sources and Searches

We systematically searched MEDLINE (via PubMed), EMBASE and Web of Science databases from their inception until 30 January 2019. The search strategy included, combined with Boolean operators, the following terms: milk, dairy, dairy product, dairy product consumption, dairy product intake, dairy intake, pulse wave velocity, arterial stiffness, PWV, arterial aging, aortic stiffness (Table 1). The literature search was complemented by reviewing the reference lists of articles considered eligible for the systematic review. Authors were contacted to obtain missing information when necessary.

### 2.2. Study Selection 

The criteria for including studies were as follows: (i) individuals from population-based samples; (ii) study design including cross-sectional studies or baseline measurements of cohort studies and RCTs; (iii) exposure to dairy products consumption, milk, cheese, yogurt, butter or other type of dairy product; and (iv) arterial stiffness as the outcome, measured using PWV. The criteria for excluding studies were as follows: (i) reports not written in English or Spanish; and (ii) non-eligible publication types, such as review articles, editorials, comments, guidelines or case-reports.

The literature search was independently conducted by two reviewers (AD-F and IC-R), and disagreements were resolved by consensus or by consultation with a third researcher (CA-B).

### 2.3. Data Extraction and Quality Assessment 

The following data were extracted from the original reports: (1) first author and year of publication; (2) country where data for the study were collected; (3) sample characteristics (type of design, sample size, average age and distribution, average systolic blood pressure (SBP), average diastolic blood pressure (DBP) and average body mass index (BMI)); (4) type of record and quantity of dairy products consumed; and (5) PWV mean value, measurement site and device used.

The Quality Assessment Tool for Observational Cohort and Cross-sectional Studies published by the National Heart, Lung and Blood Institute was used to evaluate the risk of bias of cohort and cross-sectional studies [18]. The methodological criteria included: research question, population definition, participation rate, recruitment, sample size, analysis, timeframe, exposure levels, measures and assessment, outcome measures and blinding, loss to follow-up, and confounding variables. Each study was rated as either good (i.e., most criteria met with a low risk of bias), fair (i.e., some criteria met with a moderate risk of bias), or poor (i.e., few criteria met and with a high risk of bias). Rating of studies was independently performed by two reviewers (AD-F and IC-R), and inconsistencies were resolved by consensus or by consulting a third researcher (CA-B).

### 2.4. Statistical Analysis

Both the inverse-variance fixed effects method [19] and the DerSimonian and Laird method were used to compute pooled estimates of effect size (ES) and the respective 95% confidence intervals (CI). When studies presented this association by regression models or mean value trends by group, the ES [20] was calculated. Values of ES of around 0.2 were considered a weak effect, values around 0.5 indicated a moderate effect, values around 0.8 represented a strong effect, and values larger than 1.0 were considered a very strong effect. Considering the differing composition of dairy products, the meta-analysis considered the type of dairy products, the pooled estimates by the type of dairy products were calculated only when there was three or more studies (total, milk and cheese); thus, other type of dairy products, such as cream, butter, yogurt, frozen desserts or whey were not individually analyzed. The heterogeneity of results across studies was assessed using the I2 statistic. I2 values were considered as: 0–40%, might not be important; 30–60%, may represent moderate heterogeneity; 50–90%, substantial heterogeneity; or 75–100%, considerable heterogeneity. The corresponding p-values were also taken into account [17]. 

Sensitivity analyses (systematic re-analysis while removing studies one at a time) were conducted to assess the robustness of the summary estimates and to provide insight as to whether any particular study accounted for a large proportion of heterogeneity amongst the pooled ES.

Random-effects meta-regression analyses were performed to determine whether age, BMI, SBP, DBP or pulse pressure (PP) were significant moderators for the relationship between dairy products and PWV.

Publication bias was assessed by the Harbord modified test [21], and was considered statistically significant at a p-value of <0.10. Statistical analyses were performed using Stata/SE software version 14 (College Station, TX, USA). 

## 3. Results

### 3.1. Systematic Review

The PRISMA flow diagram is presented in Figure 1. From the 208 full-text articles identified, only seven studies met the inclusion criteria and were included in the systematic review. These studies were published between 2012 and 2018, included sample sizes ranging from 22 to 12,892 participants, and data was collected from samples from Europe, North and South America and Oceania. A total of 16,427 patients met the inclusion criteria and were included in the analysis. Six studies included an adult population (mean age ranged from 26.3 ± 4.2 to 63.8 ± 12.4 years) [22,23,24,25,26,27], and one study included children and adolescent patients (13.2 ± 0.7 years) [28]. All studies determined arterial stiffness through the carotid-femoral PWV, with six studies using the SphygmoCor system (AtCor Medical Pty Ltd., West Ryde, Australia) while the other one used the Complior automatic device (Artech Medical, France) [24]. Dairy product consumption was estimated with different questionnaires that recorded the frequency and type of dairy product consumed (Table 2).

Of the studies reviewed, five focused on overweight participants [23,24,25,27,28] while the remaining two studies included obese samples [22,26]. Furthermore, five studies focused on subjects with elevated blood pressure (BP), one study focused on subjects with normal BP, and the other study focused on subjects with stage 1 hypertension [25] (Table 2). 

Six studies analyzed the consumption of total dairy products [22,23,24,25,26,27] and one only included milk intake [28]. Milk and cheese could be analyzed separately since four [24,25,26,28] and three studies [24,25,26], respectively, included the information. Butter and yogurt could not be analyzed due to the lack of results in the studies (only one study included butter [24] and two included yogurt [24,26]). 

Four studies showed negative associations between total dairy products and PWV [22,23,24,26] while two found a positive association [25,27]. Regarding milk intake, two studies found a negative association with PWV [26,28] and the other two describe a positive association [24,25]. Finally, cheese consumption was negatively associated with PWV in all included studies [24,25,26]. It is remarkable that all associations between total dairy products, milk and cheese with PWV were discrete.

Each study used a different questionnaire to assess participant’s dietary records. Among their similarities, three of them presented their results across categories of frequency of consumption of dairy products [23,24,25]. Two studies collected the data through electronic versions of the questionnaires [22,26], three collected the data at the moment of the interview [24,25,27], one required participants to fulfil a booklet every day for four days [28] and other was completed within 2 weeks of the laboratory visit by the participants [23].

Age was included as a covariate in all studies, and most of them also included as covariates anthropometric (BMI or waist circumference) [22,23,24,26,28], biochemical (high-density lipoprotein (HDL) and/or low-density lipoprotein (LDL) cholesterol, triglycerides (TG)) [23,24,27,28], the use of antihypertensive medication [23,24,25,26,27] and some blood pressure-related variables such as mean arterial pressure (MAP) or SBP (Table 2).

### 3.2. Risk of Bias

According to the Quality Assessment Tool for Observational Cohort and Cross-sectional Studies, all studies included in the meta-analysis were considered as having a low risk of bias (Table 3).

### 3.3. Meta-Analysis

Figure 2 presents the ES for each of the dairy products consumed and the PWV. Total dairy products (ES = −0.03; 95% CI [−0.04, −0.01]) and cheese (ES = −0.04; 95% CI [−0.07, −0.01]) consumption were weak but significantly associated with lower PWV levels. Conversely, milk consumption showed no significant association with PWV (ES = 0.02; 95% CI [−0.01, 0.05]). Heterogeneity in the ES was not important for the three groups of dairy products: total dairy products (I2 = 0.0%; *p* = 0.839), milk (I2 = 0.0%; *p* = 0.930), and cheese (I2 = 4.2%; *p* = 0.352).

### 3.4. Sensitivity Analysis

When the impact of individual studies was examined by removing studies from the analysis one at a time, we observed that the pooled ES estimate was modified for total dairy products (ES = −0.01, 95% CI [−0.08, 0.05]) and cheese (ES = −0.02, 95% CI [−0.10, 0.06]) consumption only after removing data from the study by Gomes-Ribeiro et al. (Table 4).

### 3.5. Meta-Regression

Due to the limited number of studies, random-effects meta-regression models were conducted only for total dairy product consumption. These analyses indicated that age (*p* = 0.983), BMI (*p* = 0.518), SBP (*p* = 0.461), DBP (*p* = 0.395) and PP (*p* = 0.512) were not related to the pooled ES estimates (Table 5).

### 3.6. Publication Bias

The Harbord modified test for publication bias was performed only for total dairy products and showed no evidence of publication bias (*p* = 0.877) (Figure 3).

## 4. Discussion

This systematic review and meta-analysis synthetized the current available evidence regarding the association between dairy product consumption and PWV. We found a significant association of total dairy and cheese consumption with slightly lower PWV levels. However, no association between milk consumption and PWV was observed. 

In our analysis, we found an overall statistically significant association between total dairy product intake and lower PWV, although the effect estimates were small. This association was mainly driven by the study by Gomez-Ribeiro et al. However, it is worth noting that, of the six studies analyzed in the category of total dairy products, only three separately analyzed the consumption of whole-fat and low-fat dairy [24,26,27]. While two studies did not find an association between PWV and whole-fat dairy, Recio-Rodriguez found opposite associations based on the fat content. Specifically, for every 100 g/day increase in low-fat dairy intake there was a decrease in PWV of 0.10 m/s (*p* = 0.011), while the same amount of high-fat dairy resulted in a similar increase in PWV (0.109 m/s; *p* = 0.038). In another cross-sectional study, Crichton et al. found that dairy product intake (80.4% of which was reduced-fat milk) was inversely associated with PWV. Low-fat dairy, but not whole-fat dairy, was associated with lower PWV in the ELSA-Brazil study [24]. Due to the cross-sectional nature of the previous results [27], these findings should be interpreted with caution. Moreover, given that data from another meta-analysis supports a negative association between total dairy product intake and CVD risk [12], together with the null association of whole-fat dairy intake reported in some studies [24,26], we suggest that conclusions for whole-fat dairy products cannot be made until we have results from well-designed follow-up or experimental studies, in accordance with all authors of the studies included in the systematic review which stated that more prospective, randomized, population-based studies are needed to reinforce their results. 

Blood pressure is one of the major determinants of arterial stiffness [36]. The various components of dairy products may explain the health benefits found in different studies. For example, the bioactive peptides present in dairy products may inhibit the angiotensin-I-converting enzyme, which is involved in the regulation of blood pressure [37,38]. Similarly, milk-derived bioactive peptides produced by the proteolysis of casein and whey have been suggested to play a role in atherosclerosis. Consistent with these biochemical mechanisms, in the prospective cohort from the Caerphilly Prospective Study, those with the highest milk intake showed a lower SBP when compared with those who did not consume milk, and milk and dairy product intake was not associated with arterial stiffening. Indeed, only butter intake was positively associated with blood pressure and arterial stiffness indicators in that study [25]. The authors highlighted that they only included whole-fat milk in their analysis. We did not include butter in this meta-analysis because from the seven studies included in this meta-analysis, only the cross-sectional ELSA-Brasil study included butter separately in their analyses [24], reporting a significant negative association between butter intake and carotid-femoral PWV, but no association with SBP and PP. Our pooled estimates did not show a significant association between milk intake and arterial stiffening; however, due to the scarcity of studies, these estimates should be interpreted with caution. 

With regard to cheese consumption, three studies included in this meta-analysis showed an overall significant association between cheese consumption and a lower PWV, although this finding was mostly driven by the results from the study by Gomes-Ribeiro et al. In their cross-sectional study, the authors reported that cheese (one serving per day) was inversely associated with PVW (cfPWV = −0.02 m/s [−0.04, −0.01]). As our results were strongly influenced by this study, we suggest that more research should address this issue. Similarly, the results of research analyzing the effect of dietary and nutrient interventions on the treatment of arterial stiffness, although limited, supports that *Lactobacillus helveticus*-fermented milk products and products containing tripeptides [39] improve arterial stiffness (Cohen’s d = 0.15–0.33) in hypertensive patients [40]. In the case of cheese consumption, because this dairy product contains a high amount of salt, further research is needed to investigate whether the specific type of cheese is relevant. Finally, yogurt was not evaluated in this meta-analysis because our search retrieved only two studies that specifically assessed its effect on arterial stiffness [24,26], although the ES estimates for these studies indicated a lower inverse association between yogurt consumption and PWV (−0.04 [−0.07, −0.01]). Thus, these results are inconclusive. Likewise, the evidence for a hypotensive effect of yogurt is controversial, as although some studies suggest a hypotensive effect [41,42], a previous meta-analysis reported moderate- to high-quality evidence that the consumption of high-fat dairy, cheese, yogurt and fermented dairy products are not associated with the risk of hypertension [42].

### Limitations

Some limitations in this systematic review and meta-analysis should be acknowledged: (1) the limited number of studies did not allow us to specifically analyze the association of each type of dairy product; (2) we could not evaluate changes or the effect of interventions on the relationship between milk intake and arterial stiffness due to the cross-sectional nature of the included studies; (3) the study conducted by Gomes-Ribeiro and colleagues had a markedly greater sample size when compared to the other studies, although sensitivity analyses did not show a change in the results after removing this study; (4) different validated food frequency questionnaires were used across studies, therefore, the data for dairy product consumption were not similar across studies which might affect the reliability of our results; and (5) although the Cochrane Handbook recommends that a meta-regression analyses only be conducted when there are 10 or more studies in the meta-analysis, we performed this statistical test for total dairy product consumption in order to obtain a picture of the impact of certain risk factors of disputed influence on PWV, such as age or blood pressure, on our estimates. Thus, the findings of these analyses should be interpreted with caution. 

## 5. Conclusions

Overall, we found no detrimental effect of dairy product consumption on arterial stiffness, as measured by PWV. An inverse association with PWV was found for total dairy and cheese consumption. However, our results should be interpreted with caution because most of the evidence comes from cross-sectional studies (seven studies). Well-designed, randomized controlled trials and follow-up studies are warranted to clarify the role of dairy products on arterial stiffness. 

## Figures and Tables

**Figure 1 nutrients-11-00741-f001:**
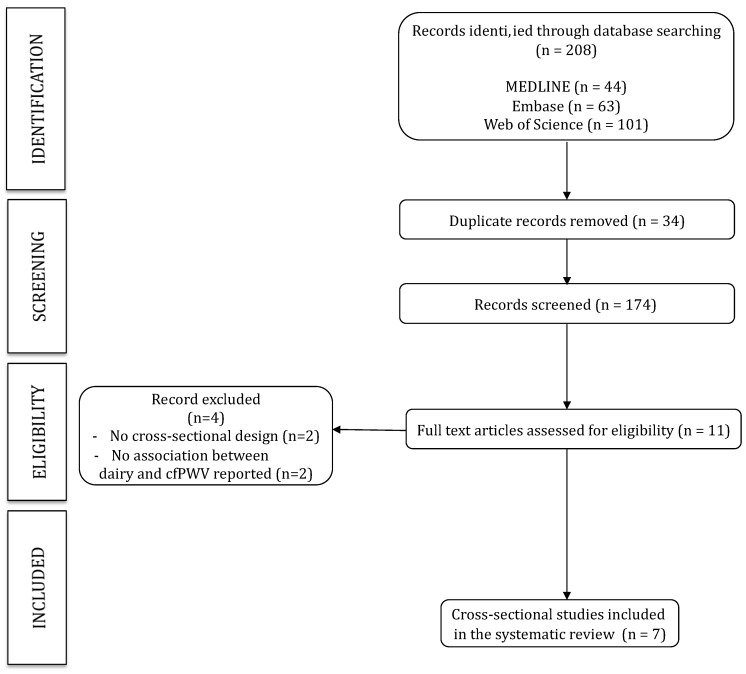
Literature search PRISMA (Preferred Reporting Items for Systematic Reviews and Meta-analyses) consort diagram.

**Figure 2 nutrients-11-00741-f002:**
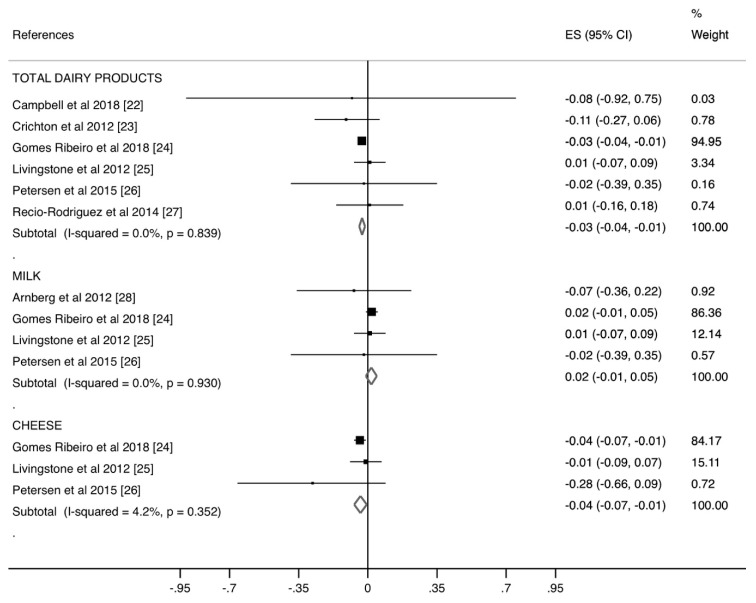
Forest plot of the estimated rate ratio of dairy products consumption and pulse wave velocity (PWV), by type of dairy product and overall.

**Figure 3 nutrients-11-00741-f003:**
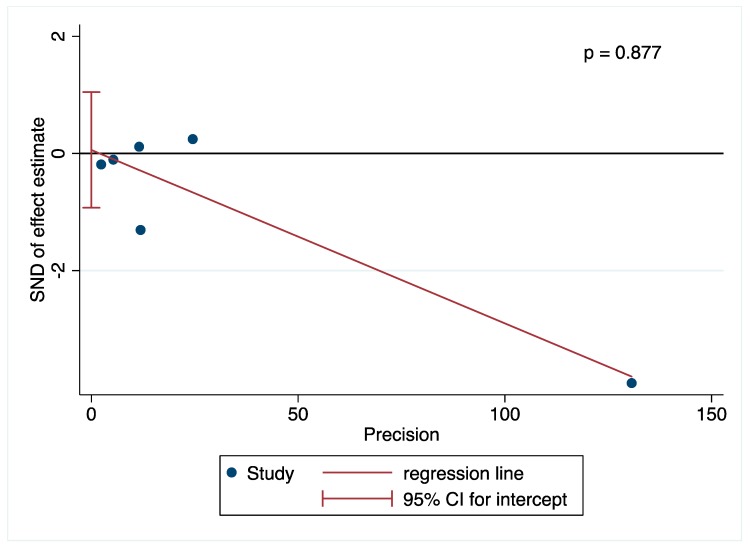
Assessment of potential publication bias by Harbord test.

**Table 1 nutrients-11-00741-t001:** Search strategy for Medline.

Dairy Related Terms		Arterial Stiffness Related Terms
“milk” OR “dairy” OR “dairy product” OR “dairy product consumption” OR “dairy product intake” OR “dairy intake”	AND	“pulse wave velocity” OR “arterial stiffness” OR “PWV” OR “arterial aging” OR “aortic stiffness”

**Table 2 nutrients-11-00741-t002:** Characteristics of the studies included.

Study	Country/Type of Design	*n*/Mean Age (Years)	Mean BMI (kg/m^2^)	Mean SBP/DBP (mmHg)	Mean PWV (m/sec)	Dietary Record	Type of Dairy Product	Mean Dairy Products Consumption	Main Results	Covariables
Arnberg K et al, 2012 [28]	Denmark/Cross-sectional	193/13.2 ± 0.7	25.2 ± 2.3	111 ± 7/65.3 ± 6.3	4.81 ± 0.71	Precoded food record of 4 days [29]	Milk	224 g/d	Negative association between milk intake and PWV.	Age, gender, MAP, Tanner stage, heart rate, HOMA, serum TG, serum HDL cholesterol and BMI.
Campbell M et al, 2018 [22]	USA/Cross-sectional	22/26.3 ± 4.2	33.2 ± 3.5	124.3 ± 8.5/79.5 ± 7.6	6.34 ± 0.88	DHQ-II of one month [30]	Dairy products	194.25 g/d	Negative correlation between dairy products and PWV.	Age, BMI, MAP, SBP, PP and waist circumference.
Crichton G et al, 2012 [23]	USA/Cross-sectional	587/63.8±12.4	29.5 ± 6.4	128.9 ± 19.6/77 ± 9.3	10.25 ± 2.8	Nutrition and Health questionnaire [31]	Milk and dairy product	NA	According to categories of dairy products frequency consumption, higher intake of dairy food was related with lower PWV, PP, and SBP values.	Age, education, sex, race, weight, heart rate, antihypertensive drug treatment, MAP, waist circumference, total cholesterol, HDL and LDL cholesterol, Center for Epidemiologic Studies Depression Scale raw score + grains per day, vegetables per day, sweets per day, protein per day, and total food servings per day.
Gomes Ribeiro A et al, 2018 [24]	Brazil/Cross-sectional	12892/51.6±8.9	26.8 ± 4.6	121.1 ± 17.1/76.3 ± 10.7	9.31 ± 1.81	Semi-quantitative FFQ [32]	Milk, cheese, yogurt, butter	316.60 g/d	Inverse association between the intake of dairy products with SBP, PP and PWV. Higher frequency of dairy product consumption was significantly associated with lower PWV values.	Age, sex, race, income, weight, height, waist circumference, smoking status, alcohol intake, physical activity, fasting glucose, total cholesterol, MAP, antidiabetic drugs (yes/no), lipid-lowering drugs (yes/no), antihypertensive drugs (yes/no), calorie intake and non-dairy food groups.
Livingstone K et al, 2013 [25]	UK/Longitudinal and cross-sectional	2373/56.9 ± 4.5	26.4 ± 3.5	144.3 ± 18.5/84.9 ± 10.1	11.47 ± 2.67	FFQ [33]	Milk, cheese, butter and cream	346 g/d	Higher intake of dairy products significantly decreased PP and SBP values, but there were not differences in PWV results across quartiles of dairy product consumption.	Age, alcohol consumption, smoking habits, social class, physical activity, total energy intake, and fat intake, heart rate, MAP, and drug use.
Petersen K et al, 2015 [26]	Australia/Cross-sectional	95/58.0±12.0	34.0 ± 6.9	129 ± 14/73 ± 10	9.6 ± 1.8	DQES v2 FFQ of 3 days [34]	Milk, yogurt and cheese	387 g/d	Significant inverse association between total dairy intake and yogurt with PWV.	Age, central MAP, BMI, heart rate and antihypertensive medication prescription.
Recio-Rodriguez JI et al 2014 [27]	Spain/Cross-sectional	265/55.9±12.2	27.3 ± 4.3	122.3 ± 17.9/77.6 ± 10.8	7.60 ± 2.00	Semi-quantitative 137-item FFQ [35]	High-fat dairy and low-fat dairy	133.1 g/d	Low-fat dairy products consumption was associated with lower PWV values, but whole-fat products increased PWV values.	Age, sex, BMI, smoking, SBP, total cholesterol, energy intake and the presence of diabetes, antihypertensive, antidiabetic and lipid-lowering drugs.

BMI: Body mass Index; SBP: Systolic Blood Pressure; DBP: Diastolic Blood Pressure; PWV: Pulse Wave Velocity; m/sec: meters/second; FFQ: Food-frequency questionnaire. NA: Not available.

**Table 3 nutrients-11-00741-t003:** Quality assessment tool for observational cohort and cross-sectional studies.

	Arnberg K et al, 2012	Campbell M et al, 2018	Crichton G et al, 2012	Gomes Ribeiro A et al, 2018	Livingstone K et al, 2013	Petersen K et al, 2015	Recio-Rodriguez JI et al, 2014
1. Was the research question or objective in this paper clearly stated?	Yes	Yes	Yes	Yes	Yes	Yes	Yes
2. Was the study population clearly specified and defined?	Yes	Yes	Yes	Yes	Yes	Yes	Yes
3. Was the participation rate of eligible persons at least 50%?	Yes	NA	Yes	Yes	Yes	Yes	NA
4. Were all the subjects selected or recruited from the same or similar populations (including the same time period)? Were inclusion and exclusion criteria for being in the study prespecified and applied uniformly to all participants?	Yes	Yes	Yes	Yes	Yes	Yes	Yes
5. Was a sample size justification, power description, or variance and effect estimates provided?	No	No	No	Yes	No	No	Yes
6. For the analyses in this paper, were the exposure(s) of interest measured prior to the outcome(s) being measured?	No	No	No	No	No	No	No
7. Was the timeframe sufficient so that one could reasonably expect to see an association between exposure and outcome if it existed?	No	No	No	No	No	No	No
8. For exposures that can vary in amount or level, did the study examine different levels of the exposure as related to the outcome (e.g., categories of exposure, or exposure measured as continuous variable)?	No	No	Yes	Yes	Yes	No	No
9. Were the exposure measures (independent variables) clearly defined, valid, reliable, and implemented consistently across all study participants?	Yes	Yes	Yes	Yes	Yes	Yes	Yes
10. Was the exposure(s) assessed more than once over time?	NA	NA	NA	NA	NA	NA	NA
11. Were the outcome measures (dependent variables) clearly defined, valid, reliable, and implemented consistently across all study participants?	Yes	Yes	Yes	Yes	Yes	Yes	Yes
12. Were the outcome assessors blinded to the exposure status of participants?	Yes	Yes	Yes	Yes	Yes	Yes	Yes
13. Was loss to follow-up after baseline 20% or less?	NA	NA	NA	NA	NA	NA	NA
14. Were key potential confounding variables measured and adjusted statistically for their impact on the relationship between exposure(s) and outcome(s)?	Yes	Yes	Yes	Yes	Yes	Yes	Yes

NA: Not applicable.

**Table 4 nutrients-11-00741-t004:** Sensitivity analysis by removing one by one the included studies.

References	ES (95% CI)	*I* ^2^	*p*
**TOTAL DAIRY PRODUCTS**			
Campbell M et al, 2018 [22]	−0.03 (−0.04, −0.01)	0.0	0.725
Crichton G et al, 2012 [23]	−0.03 (−0.04, −0.01)	0.0	0.888
Gomes Ribeiro A et al, 2018 [24]	−0.01 (−0.08, 0.05)	0.0	0.785
Livingstone K et al, 2013 [25]	−0.03 (−0.05, −0.02)	0.0	0.889
Petersen K et al, 2015 [26]	−0.03 (−0.04, −0.01)	0.0	0.722
Recio-Rodríguez JI et al, 2014 [27]	−0.03 (−0.04, −0.01)	0.0	0.759
**MILK**			
Arnberg K et al, 2012 [28]	0.02 (−0.01, 0.05)	0.0	0.954
Gomes Ribeiro A et al, 2018 [24]	0.00 (−0.07, 0.08)	0.0	0.866
Livingstone K et al, 2013 [25]	0.02 (−0.01, 0.05)	0.0	0.815
Petersen K et al, 2015 [26]	0.02 (−0.01, 0.05)	0.0	0.815
**CHEESE**			
Gomes Ribeiro A et al, 2018 [24]	−0.08 (−0.31, 0.15)	47.5	0.168
Livingstone K et al, 2013 [25]	−0.04 (−0.07, −0.01)	36.0	0.211
Petersen K et al, 2015 [26]	−0.04 (−0.06, −0.01)	0.0	0.815

**Table 5 nutrients-11-00741-t005:** Meta-regressions of the PWV and total dairy products consumption by age, body mass index (BMI), systolic blood pressure (SBP), diastolic blood pressure (DBP) and pulse pressure (PP) of included studies.

Covariates	ß (95% CI)	*p*
Age (years)	−0.00 (−0.01, 0.01)	0.983
BMI (kg/m^2^)	−0.01 (−0.07, 0.04)	0.518
SBP (mmHg)	0.00 (−0.00, 0.01)	0.461
DBP (mmHg)	0.00 (−0.01, 0.02)	0.395
PP (mmHg)	0.00 (−0.01, 0.01)	0.512

SBP: Systolic blood pressure; DBP: Diastolic blood pressure; PP: Pulse pressure.

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
