# Peer review of "Total Dairy, Cheese and Milk Intake and Arterial Stiffness: A Systematic Review and Meta-analysis of Cross-Sectional Studies"

_nutrients, 2019, doi:10.3390/nu11040741_

Round 1

Reviewer 1 Report

Overall, the review manuscript by Diez-Fernandez is well written. However, as the authors stated, the manuscript has several limitations, particularly, the small number of studies were included because of strict study selection criteria. Besides, some minor points need to be addressed. 

The study focused on ‘total dairy, cheese, and milk’ instead of all dairy products including ‘milk, cream, butter, cheese, yogurt, frozen desserts and whey’. The title is inappropriate and should be changed. 

In the line 107, ‘the inverse-variance fixed effects method and the DerSimonian and Laird method’ should be clearly described. The cut-offs defined for  week effect, moderate effect, strong effect, and very strong effect are 0.2, 0.5, 0.8, and 1.0, respectively. But the authors stated in the abstract and ‘3.3. Meta-Analysis’ that total dairy products (ES = -0.03) and cheese (ES = -0.04) were significant. Please clarify this.  

Line 145-147, Figure 1 has format errors. It needs to be changed. 

In sections 3.3, 3.4, 3.5, 3.6, all data are not shown. The data need to be put together into a Table.

Line 169, change ‘3.5 Publication bias’ to ‘3.6 Publication bias’

Author Response

Comments to Reviewer 1

Overall, the review manuscript by Diez-Fernandez is well written. However, as the authors stated, the manuscript has several limitations, particularly, the small number of studies were included because of strict study selection criteria. Besides, some minor points need to be addressed. 

Reviewer:The study focused on ‘total dairy, cheese, and milk’ instead of all dairy products including ‘milk, cream, butter, cheese, yogurt, frozen desserts and whey’. The title is inappropriate and should be changed. 

Authors:Thank you for the reviewer’s comment. We have modified the title as suggested.

Reviewer:In the line 107, ‘the inverse-variance fixed effects method and the DerSimonian and Laird method’ should be clearly described. The cut-offs defined for  week effect, moderate effect, strong effect, and very strong effect are 0.2, 0.5, 0.8, and 1.0, respectively. But the authors stated in the abstract and ‘3.3. Meta-Analysis’ that total dairy products (ES = -0.03) and cheese (ES = -0.04) were significant. Please clarify this.  

Authors: We agree with the reviewer. We have modified both sentences as follows:

“Total dairy product (ES = -0.03; 95% CI [-0.04, -0.01]) and cheese (ES = -0.04; 95% CI [-0.07, -0.01]) consumption were weak butsignificantly associated with lower PWV levels.”

Reviewer:Line 145-147, Figure 1 has format errors. It needs to be changed. 

Authors:We apologize for the inconvenience. The figure has been uploaded separately of the manuscript with enough quality. 

Reviewer:In sections 3.3, 3.4, 3.5, 3.6, all data are not shown. The data need to be put together into a Table.

Authors:We agree with the reviewer’s comment. Data have been added as supplementary material.

Reviewer:Line 169, change ‘3.5 Publication bias’ to ‘3.6 Publication bias’

Authors: The subheading has been corrected. Thank you.

Reviewer 2 Report

Although the relationship between dairy products and cardiometabolic diseases is not new and some systematic reviews and meta-analysis have been previously published, the authors have made a significant contribution to the knowledge. They addressed the searching of the literature related to arterial stiffness as study outcome and a marker of CVD events. The aim of the study is clearly justified, the study is clearly described and manuscript is well organized via all sections, but some details should be improved.

My key remark is related to the justification of the type of dairy foods that were chosen to study association with outcome – now the justification is given in the discussion section, in my opinion such description should be given in the methods section.

Detailed comments

Title: I suggest completing the title by adding  type of the study (cross-sectional) related to the review

Lines 22-24, 76-77: The search of databases should be determined exactly according to the date  from day/month/year  to day/month/year.

Lines 42-44: The Authors described ‘a negative association between dairy product intake and the risk of CVD…’ but also other findings have been reported, some evidence showed no association. The first paragraph should report more studies including opposite/neutral findings.

Lines 114-116: Some readers may be confused by reading ranges of I2 values that overlap, e.g. 0-40%; 30-60%; etc.

Fig.1 should the improved – now is unreadable.

Author Response

Comments to Reviewer 2

Although the relationship between dairy products and cardiometabolic diseases is not new and some systematic reviews and meta-analysis have been previously published, the authors have made a significant contribution to the knowledge. They addressed the searching of the literature related to arterial stiffness as study outcome and a marker of CVD events. The aim of the study is clearly justified, the study is clearly described and manuscript is well organized via all sections, but some details should be improved.

Reviewer:My key remark is related to the justification of the type of dairy foods that were chosen to study association with outcome – now the justification is given in the discussion section, in my opinion such description should be given in the methods section.

Authors:Thank you, we have added a sentence in the Statistical analysis subsection of the Methods section:

“Considering the differing composition of dairy products, the pooled estimates by the type of dairy products were calculated only when there was three or more studies (total, milk and cheese); thus, other type of dairy products, such as cream, butter, yogurt, frozen desserts orwheywere not individually analyzed.”

Detailed comments

Reviewer:Title: I suggest completing the title by adding  type of the study (cross-sectional) related to the review.

Authors:Thank you for the reviewer’s comment. We have modified the title as suggested.

Reviewer:Lines 22-24, 76-77: The search of databases should be determined exactly according to the date  from day/month/year  to day/month/year.

Authors:Thank you, the full date has been added.

Reviewer:Lines 42-44: The Authors described ‘a negative association between dairy product intake and the risk of CVD…’ but also other findings have been reported, some evidence showed no association. The first paragraph should report more studies including opposite/neutral findings.

Authors:Thank you for the reviewer’s comment. A sentence has been included regarding  neutral effects and two references have been added in the third paragraph regarding the increased risk of CVD: 

“Moreover, a recent meta-analysis of prospective studies have also shown a null or weak inverse effect of dairy products on CVD [5].”

“Indeed, the SFA content is the main reason why dairy product intake has received an unfavorable reputation in recent years [10].

Reviewer:Lines 114-116: Some readers may be confused by reading ranges of I2 values that overlap, e.g. 0-40%; 30-60%; etc.

Authors:Thank you for the reviewer suggestion. We have followed the Cochrane Collaboration Handbook in which is stated that establishing thresholds for the interpretation of I2 can be confusing, as the importance of inconsistency depends on several factors, but since there are no levels of heterogeneity in the paper that could lead to misinterpretation, we prefer to show the interpretation as published in the Handbook.

Reviewer: Fig.1 should the improved – now is unreadable.

Authors:We apologize for the inconvenience. The figure has been uploaded separately of the manuscript with enough quality.